# Assessing Social and Intersectional Biases in Contextualized Word Representations

**Yi Chern Tan, L. Elisa Celis**
Yale University
{yichern.tan, elisa.celis}@yale.edu

## Abstract

Social bias in machine learning has drawn significant attention, with work ranging from demonstrations of bias in a multitude of applications, curating definitions of fairness for different contexts, to developing algorithms to mitigate bias. In natural language processing, gender bias has been shown to exist in context-free word embeddings. Recently, contextual word representations have outperformed word embeddings in several downstream NLP tasks. These word representations are conditioned on their context within a sentence, and can also be used to encode the entire sentence. In this paper, we analyze the extent to which state-of-the-art models for contextual word representations, such as BERT and GPT-2, encode biases with respect to gender, race, and intersectional identities. Towards this, we propose assessing bias at the contextual word level. This novel approach captures the contextual effects of bias missing in context-free word embeddings, yet avoids confounding effects that underestimate bias at the sentence encoding level. We demonstrate evidence of bias at the corpus level, find varying evidence of bias in embedding association tests, show in particular that racial bias is strongly encoded in contextual word models, and observe that bias effects for intersectional minorities are exacerbated beyond their constituent minority identities. Further, evaluating bias effects at the contextual word level captures biases that are not captured at the sentence level, confirming the need for our novel approach.

## 1 Introduction

Word embeddings [22, 24], which provide context-free vector representations of words, have become standard practice in NLP. Recently, contextual word representations [19, 17, 25, 26, 10, 27] have had significant success in improving downstream NLP tasks, and most state-of-the-art systems use such representations of words. These models rely on a pre-trained encoder network run on a sentence to output contextual embeddings for each token. The networks can be based on Long Short-Term Memory (LSTM) units or a transformer, and the corresponding embeddings can be used in the same manner as context-free word embeddings.

However, word embeddings such as word2vec [22] and GloVe [24] have been shown to exhibit social biases, including gender bias [2, 5] and racial bias [20]. These biases are concerning, as word embeddings form the foundation of most language systems. Their use can perpetuate and even amplify cultural stereotypes by encoding them into language systems. Downstream tasks such as classification and ranking will suffer from the implicit encoding of these biases, posing a hurdle to building fair machine learning systems [31]. In this work we extend and expand on the analysis of contextual representations with respect to social and intersectional biases. Zhao et al. [35] demonstrate that ELMo [25] contextual word representations exhibit gender bias. However, BERT has since surpassed ELMo in performance on downstream NLP tasks like natural language understanding, inference, question answering, and named entity recognition [10]. BERT is composed of a layered self-attention

transformer network trained on Wikipedia and the BookCorpus dataset [36]. Furthermore, GPT-2, which is another transformer-based model trained on the WebText dataset, has also set state-of-the-art benchmarks [27]. It is thus imperative to evaluate the extent to which they exhibit social and intersectional bias.

May et al. [21] establish a preliminary study of social bias in BERT, but their analysis relies only on sentence level encodings. Our work extends beyond this to provide an analysis of gender and racial bias on a variety of state-of-the-art contextual word models. We also evaluate intersectional (gender + race) bias, since the lived experience of groups with multiple minority identities is cumulatively worse than that of each of the groups with a singular minority identity [9]. We adapt the Sentence Encoder Association Test (SEAT) [21] to evaluate how these techniques displays bias in contextual word representations. This effectively gives a new metric that considers a word embedding and its bias *in context*.

We find that the standard corpora for pre-training contextual word models exhibit significant gender bias imbalances. Furthermore, we show evidence of social and intersectional bias in state-of-the-art contextual word models. In addition, the biases are detected at different levels and in different instances in contextual word representations as compared to sentence-level representations. This suggests both encoding types are needed to measure bias as they capture context in different ways, and any proposed de-biasing techniques should consider both metrics in their evaluation. We also show that racial bias is encoded strongly in contextual word models, perhaps more so than gender bias. Lastly, we introduce a method of comparison to provide evidence that intersectional identities (in this study, African American females) suffer from such biases as well, more so than their constituent minority identities, and that the effect of race seems to be larger than the effect of gender for such intersectional identities.

## 2   Related Work

Social bias in language has been demonstrated to exist upstream in a variety of corpora and datasets; on a corpus elicited from crowdworkers [28], and on the 1 Billion Word Benchmark corpus [7] where it was observed that there was gender skew in proportions of gendered pronouns and associations with occupation words Zhao et al. [35]. Gender bias has also been demonstrated downstream on several applications of natural language processing, including sentiment analysis [18, 30], abusive language detection [23], image captioning [16] and text classification [11]. These models not only reflect the bias in training data, but also amplify the bias [32, 5]. Our work extends the upstream corpus level analysis in two ways: 1) we include the non-gendered or collective pronoun in occurrence counts, and 2) we apply the analysis on other datasets like BooksCorpus [36], Wikipedia, and WebText [27].

Significant work has been done to show social bias (in particular gender bias) in word embeddings. Bolukbasi et al. [2] and Caliskan et al. [5] demonstrate that word embeddings associate occupations with their stereotypical gender roles (eg. doctors are stereotypically male, nurses are stereotypically female) by evaluating occupation words with Word Embedding Association Tests (WEATs) to gender words. Inspired by implicit association tests, WEATs compute the differences in distances when word vectors are asked to pair two concepts that are similar (e.g., stereotypically female occupation words and female gender words) as opposed to two concepts that are different (e.g., stereotypically male occupation words and female gender words). Brunet et al. [4] trace the internalization of gender bias to representational differences at the corpus level. This line of work has very recently been extended to evaluate gender bias in contextual word representations in some specific settings. Zhao et al. [35] and Basta et al. [1] demonstrate gender bias in ELMo [25] word embeddings, whereas May et al. [21] evaluate various models of contextual word representations on a sentential generalization of WEAT. Our work extends such analyses in two ways: 1) we consider a wide variety of contextual word embedding tools including state-of-the-art approaches such as BERT and GPT-2, 2) we extend the evaluation to consider *contextual* word representations as opposed to prior work, which either used word embeddings without context or used sentence-level embeddings that can have additional confounds and obscure bias, and 3) we include additional embedding association tests targeting gender, race and intersectional identities.

**Table 1:** Occurrence statistics for the 1 Billion Word Benchmark, BookCorpus, Wikipedia, and WebText datasets. We show counts for gendered and neutral pronouns, co-occurrence with stereotypically male occupation words (M-biased), co-occurrence with stereotypically female occupation words (F-biased), and co-occurrences with neutral words (the rest).

|  |  | #occcurrence | M-biased occs (%) | F-biased occs (%) | Neutral (%) |
|---|---|---|---|---|---|
| **1BWord** | he/him/his | 5370000 | 3.40 | 1.52 | 95.09 |
|  | she/her/hers | 1620000 | 2.12 | 2.28 | 95.60 |
|  | they/them/their/theirs | 3890000 | 1.89 | 0.88 | 97.23 |
| **BooksCorpus** | he/him/his | 18700000 | 0.51 | 0.24 | 99.25 |
|  | she/her/hers | 13500000 | 0.32 | 0.28 | 99.40 |
|  | they/them/their/theirs | 5330000 | 0.52 | 0.23 | 99.25 |
| **Wikipedia** | he/him/his | 16300000 | 3.26 | 3.39 | 93.35 |
|  | she/her/hers | 4820000 | 1.90 | 3.39 | 94.71 |
|  | they/them/their/theirs | 7510000 | 1.78 | 1.25 | 96.97 |
| **WebText** | he/him/his | 299000 | 2.35 | 1.13 | 96.52 |
|  | she/her/hers | 95500 | 1.55 | 1.74 | 96.72 |
|  | they/them/their/theirs | 261000 | 1.22 | 0.57 | 98.21 |

## 3 Gender Bias in Datasets

### 3.1 Counting Occurrences in Datasets

BERT [10] was trained on Wikipedia (2,500M words) [1] and BooksCorpus (800M words) [36]. ELMo [25] was trained on the 1 Billion Word Benchmark (1,000M words) [7]. GPT [26] was trained on BooksCorpus, and GPT-2 was trained on WebText [27]. We follow Zhao et al. [35] in counting, in these datasets, the occurrence of a male pronoun or a female pronoun in a sentence, as well as the co-occurrences of stereotypically gendered occupation words with the respective pronouns in a sentence. We depart from previous work by including the nominative (she), accusative (her), prenominal possessive (her) and predicative possessive (hers) inflections of personal pronouns, and we also include the non-gendered or collective pronoun they. Specifically, for each sentence, we increment a count for female pronoun occurrence if any female pronoun is in the sentence, and then count the number pro-stereotypical and anti-stereotypical associations with occupation words (same for male and non-gendered pronouns). Pro-stereotypical associations are referred to as occurrences of the noun phrase referring to a profession that is male-dominated and is linked to a male pronoun, or female-dominated linked to a female pronoun (anti-stereotypical associations are the opposite). The occupation words were obtained from the WinoBias dataset [33], which were gathered from the US Department of Labor. Since the WebText dataset has not been released fully by [27], we only use the partially released version with 250K documents [2].

### 3.2 Analysis of Occurrence Statistics

First, in all the examined datasets, the occurrence of male pronouns is consistently higher than female pronouns. On the BooksCorpus dataset, this factor is only 1.3x, whereas on the 1 Billion Word Benchmark, Wikipedia and WebText, this factor is 3x. Similarly, on the BooksCorpus dataset, the non-gendered or collective pronoun they and its inflections occur the least frequently, but on the other datasets they occur second in frequency to the male pronouns. This could be because fiction books tend to have greater parity in gender representation and less references to non-gendered or collective pronouns, but online text data may have greater representation of male and non-gendered or collective entities. Second, we also observe that on the 1 Billion Word Benchmark and WebText datasets, the occurrences of pro-stereotypical associations for both male and female pronouns are proportionally higher than the anti-stereotypical associations, suggesting evidence of gender bias in these datasets. On the BooksCorpus dataset, stereotypically male occupation words co-occur proportionally more than stereotypically female occupation words with gendered personal pronouns, regardless of the gender. Conversely, on the Wikipedia dataset, the inverse relation is true. Third, non-gendered or collective pronouns co-occur more frequently with stereotypically male occupation words across all datasets,

suggesting once again an implicit bias in these datasets. The full results are shown in Table 1. In Table 3 and section 5.3, we use the embedding association test (described below) to understand the bias of the respective contextual word models given their datasets by analyzing their association of female and male names with occupation words.

We note that the demonstration of bias in these datasets is a reflection of social bias captured in language. Furthermore, the use of such datasets is entrenched into NLP practice due to the high costs of constructing new datasets and importance of dataset size in training models, particularly contextual word models. Barring the precise construction of datasets that reach representational parity across social groups, other techniques like corpus-level constraints [32], learning constraints [6] or post-processing [2] are likely to be more convenient solutions, though they ignore the origins of the problem [4].

## 4 Social and Intersectional Bias using Embedding Association Tests

### 4.1 Embedding Association Tests

We adopt the methodology of Caliskan et al. [5] and May et al. [21] to test social and intersectional bias using embedding association tests with contextual word representations. Caliskan et al. [5] proposed Word Embedding Association Tests (WEATs), which follows human implicit association tests [14] in measuring the association between two target concepts and two attributes.

We follow May et al. [21] in describing WEATs and SEATs. Let $X$ and $Y$ be equal-size sets of target concept embeddings, and $A$ and $B$ be sets of attribute embeddings. These embeddings are obtained after encoding a set of words which define the concept or attribute. Intuitively, WEATs measure the effect size of the association between a concept $X$ with attribute $A$ and concept $Y$ with attribute $B$, as opposed to concept $X$ with attribute $B$ and concept $Y$ with attribute $A$. The test statistic is

$$s(X, Y, A, B) = \sum_{x \in X} s(x, A, B) - \sum_{y \in Y} s(y, A, B) \tag{1}$$

where each addend is the difference between the mean of cosine similarities of the respective attributes:

$$s(w, A, B) = \text{mean}_{a \in A} \cos(w, a) - \text{mean}_{b \in B} \cos(w, b). \tag{2}$$

To compute the significance of the association between $(A, B)$ and $(X, Y)$, a permutation test on $s(X, Y, A, B)$ is used.

$$p = \Pr[s(X_i, Y_i, A, B) > s(X, Y, A, B)] \tag{3}$$

where the probability is computed over the space of partitions $(X_i, Y_i)$ of $X \cup Y$ so that $X_i$ and $Y_i$ are of equal size. The effect size is defined to be

$$d = \frac{\text{mean}_{x \in X} s(x, A, B) - \text{mean}_{y \in Y} s(y, A, B)}{\text{std\_dev}_{w \in X \cup Y} s(w, A, B)}. \tag{4}$$

A larger effect size corresponds to more severe pro-stereotypical representations, controlling for significance.

May et al. [21] adopt the WEAT tests [5] into Sentence Encoder Association Tests (SEATs) to test biases using sentence encodings. The embeddings used in the association tests are encodings of a sentence, which are obtained by pooling per token contextual representations, or by using the representation of the first or last token. SEATs are constructed from WEATs by using "semantically bleached" sentence templates such as "This is a [doctor]" or "[Alice] is here". These "semantically bleached" templates were created to observe the effect of a sentence encoding based on a given term, instead of the associations made with the context of other potentially semantically meaningful words. We refer to WEATs and SEATs as Caliskan tests [5].

**Table 2:** Proportion of significant positive effect sizes across embedding association tests, broken down by type of identity and model. Significance level of 0.01. Other embedding association tests were conducted (C9: disability, C10: age) but are reported only in the Supplementary Material. The total number of embedding association tests was 92: 34 gender, 31 race, 21 intersectional, 6 disability, age. For CBoW the c-word encoding tests are invalid, so the numbers are 22, 20, 14, 4 respectively.

| Test | CBoW | ELMo | BERT (bbc) | BERT (blc) | GPT | GPT-2 (117M) | GPT-2 (345M) |
|---|---|---|---|---|---|---|---|
| gender | 0.73 | 0.03 | 0.32 | 0.12 | 0.35 | 0.24 | 0.15 |
| race | 0.60 | 0.10 | 0.58 | 0.58 | 0.39 | 0.42 | 0.42 |
| intersectional | 0.29 | 0.10 | 0.71 | 0.38 | 0.33 | 0.29 | 0.10 |
| disability, age | 0.75 | 0.17 | 0.00 | 0.00 | 0.17 | 0.33 | 0.17 |
| **Overall** | 0.58 | 0.08 | 0.48 | 0.33 | 0.35 | 0.32 | 0.23 |

## 4.2 Extension of Embedding Association Tests to Contextual Word Representations

May et al. [21] suggest that although they find less bias in sentence encoders than context free word embeddings, the sentence templates may not be as semantically bleached as expected, and that a lack of evidence of bias should not be taken as a lack of bias. We propose to assess bias at the contextual word level. This allows an investigation into the bias of contextual word representation models (which allow for sentence encoding), and at the same time avoids confounding contextual effects at the sentence level, which can obscure bias.

To determine underlying bias in contextual word representations, we adopt SEATs and make a simple modification. Instead of using the sentence encoding for the association tests, we use the contextual word representation of the token of interest (i.e. we use the representation of the word before it is pooled). For example, in BERT the sentence encoding is obtained as the representation of the `[CLS]` token; in GPT it is the representation of the last token; in ELMo it is obtained by mean-pooling over all token representations. However, in all cases we instead use the contextual word encoding corresponding to the token representation of interest. More precise implementation details are in the Supplementary Material.

## 4.3 New Embedding Association Tests for Social and Intersectional Biases

To investigate social and intersectional bias, we introduce new embedding association tests to more comprehensively target race, gender and intersectional identities. The new tests are prefixed by a "+" in Tables 3, 4 and 5. For race and gender, we are interested in attributes of pleasantness (P/U: Pleasant/Unpleasant), work (Career/Family), discipline (Science/Arts) [5] and the Heilman double bind [15, 21]. The Heilman double bind refers to how women, when clearly succeeding in a stereotypically male occupation, are perceived as less likable than similar men, and how women, when success is more ambiguous, are perceived as less competent than similar men. Although the Heilman double bind originated in the context of gender, we also extend the attribute lists [3] of competence and likability to the context of race. We preserve and report the original WEATs, SEATs and tests introduced by May et al. [21] where possible. We also prefer tests using names (e.g. Alice) as concept words over group terms (e.g. European American), since names were demonstrated to have a significant association more often than group terms [21] [4]. Specifically, for both the new gender tests (+C11, +Occ) and new race tests (+C12, +C13, +Double Bind), we use the male, female, European American or African American names from existing tests and match them with the appropriate attribute words from existing tests [5] For example, test +C11 was created by matching male and female names from test C6 with attribute words of pleasantness from test C3. The new double bind tests were created by matching European American and African American names from test C3 with attribute words in existing double bind tests. More information on how the tests were created can be found in the Supplementary Material.

For intersectional identities, we are focused primarily on the identity which is the subject of discussion in the work of Crenshaw [9]: being both African American and female. Specifically,

**Table 3:** Gender embedding association tests and effect sizes, for word encodings (word), sentence encoding (sent) and contextual word representation (c-word). M/F: Male/Female. P/U: Pleasant/Unpleasant. sent(u)/c-word(u): unbleached sentence templates were used. Tests introduced in this paper are prefixed by "+". Gray shading indicates significant at $p = 0.01$. Green check (✅): test is significant using c-word but not sent. Red cross (❌): test is significant using sent but not c-word. Yellow triangle (⚠️): test is significant using both sent and c-word.

| Test | Encoding | CBoW | ELMo | BERT (bbc) | BERT (blc) | GPT | GPT-2 (117M) | GPT-2 (345M) |
|---|---|---|---|---|---|---|---|---|
| +C11: M/F Names, P/U | word | -1.31 | +0.34 | +0.69 | +0.83 | -0.43 | +0.82 | -0.10 |
| +C11: M/F Names, P/U | sent | -0.87 | +0.15 | +0.68 ❌ | +0.18 | -0.64 | +0.27 | -0.17 |
| +C11: M/F Names, P/U | c-word | NA | +0.14 | -0.44 | +0.27 | -0.35 | +0.46 ✅ | -0.13 |
| C6: M/F Names, Career/Family | word | +1.81 | -0.44 | -0.49 | -0.51 | -0.10 | -0.25 | -0.27 |
| C6: M/F Names, Career/Family | sent | +1.74 | -0.38 | -0.74 | -0.57 | +1.04 | +0.27 | +0.25 |
| C6: M/F Names, Career/Family | c-word | NA | -0.10 | +0.67 ✅ | -0.04 | +1.07 ⚠️ | +0.39 ✅ | -0.26 |
| C8: Science/Arts, M/F Terms | word | +1.24 | +0.24 | -0.23 | -0.15 | +0.25 | +0.51 | +0.87 |
| C8: Science/Arts, M/F Terms | sent | +1.01 | -0.30 | +0.11 | -0.16 | +0.89 | -0.15 | -0.15 |
| C8: Science/Arts, M/F Terms | c-word | NA | +0.16 | +1.02 ✅ | -0.08 | +1.03 ⚠️ | +0.64 ✅ | +0.67 ✅ |
| Double Bind M/F (Competent) | word | +1.62 | -0.34 | -0.35 | -0.26 | -0.66 | +1.00 | -0.04 |
| Double Bind M/F (Competent) | sent | +0.79 | -0.15 | -0.06 | 0.00 | +0.27 | +0.52 ❌ | +0.25 |
| Double Bind M/F (Competent) | c-word | NA | -0.07 | +0.42 ✅ | +0.02 | -0.02 | -0.94 | +0.57 ✅ |
| Double Bind M/F (Competent) | sent (u) | +0.84 | +0.21 | +0.39 | +0.60 | -0.76 | +1.26 ❌ | -0.59 |
| Double Bind M/F (Competent) | c-word (u) | NA | -0.48 | +0.46 | -0.37 | -0.36 | -0.72 | +0.56 |
| Double Bind M/F (Likable) | word | +1.29 | -0.61 | -1.37 | -0.64 | +0.15 | +0.83 | +0.02 |
| Double Bind M/F (Likable) | sent | +0.69 | -0.45 | -0.66 | -0.29 | -0.53 | -0.44 | -0.13 |
| Double Bind M/F (Likable) | c-word | NA | -0.38 | +0.64 ✅ | +0.13 | -0.03 | -0.68 | +0.50 ✅ |
| Double Bind M/F (Likable) | sent (u) | +0.51 | -0.92 | +0.74 | -0.97 | -1.57 | +0.25 | -1.01 |
| Double Bind M/F (Likable) | c-word (u) | NA | +0.20 | +1.29 ✅ | -0.78 | -1.22 | -0.98 | +0.39 |
| +Occ: M/F Names, Occ Terms | word | +1.59 | +0.63 | +0.55 | +0.65 | -0.38 | +0.76 | +0.46 |
| +Occ: M/F Names, Occ Terms | sent | +1.48 | +0.06 | +0.30 | +0.51 | +1.74 ❌ | -0.00 | -0.27 |
| +Occ: M/F Names, Occ Terms | c-word | NA | -0.27 | +0.98 ⚠️ | +0.67 ⚠️ | +0.10 | +0.27 ✅ | +0.43 ✅ |

we anchor comparison in the most privileged group `EuropeanAmerican+male`, and compare against `AfricanAmerican+male` (test +I3) and `EuropeanAmerican+female` (test +I4), and finally the group with identity that intersects both less privileged axes of gender and race, `AfricanAmerican+female` (test +I5). This allows us to compare the effect of being both African American and female, relative to being African American or being female. Moreover, we provide further tests where the anchoring comparison is respect to `AfricanAmerican+female`, and compare against `EuropeanAmerican+female` (test +I1) and `AfricanAmerican+male` (test +I2), and the most privileged group `EuropeanAmerican+male` (test +I1). The new tests were created by matching names from existing tests with the attribute words of pleasantness from test C3. We also report the Angry Black Woman Stereotype test introduced by May et al. [21], which targets the stereotype of black women as loud, angry, and imposing [8].

## 5 Empirical Analysis

### 5.1 Experiments

We investigate biases in GPT-2 [27], one of the state-of-the-art models for contextual word representations, in both its 117M and 345M versions. For comparison with previous work [1, 21, 35], we also report on other word representation models: CBoW-GLoVe [24], ELMo [25], BERT `bert-base-cased` (bbc) and `bert-large-cased` (blc) versions [10], and GPT [26]. For all association tests, we use $p = 0.01$ for significance testing. We use PyTorch, as well as the framework and code from May et al. [21], to conduct the experiments [6].

### 5.2 Overall Analysis

We report the proportion of tests with significant effects in Table 2. Note that all instances of significant effects had positive effect sizes. We observe that the context-free GloVe model exhibits the highest

**Table 4:** Race embedding association tests and effect sizes, for word encodings (word), sentence encoding (sent) and contextual word representation (c-word). EA/AA: European American/African American. P/U: Pleasant/Unpleasant. sent(u)/c-word(u): unbleached sentence templates were used. Tests introduced in this paper are prefixed by "+". Gray shading indicates significant at $p = 0.01$. Green check (✅): test is significant using c-word but not sent. Red cross (❌): test is significant using sent but not c-word. Yellow triangle (⚠️): test is significant using both sent and c-word.

| Test | Encoding | CBoW | ELMo | BERT (bbc) | BERT (blc) | GPT | GPT-2 (117M) | GPT-2 (345M) |
|---|---|---|---|---|---|---|---|---|
| C3: EA/AA Names, P/U | word | +1.41 | -0.41 | +0.38 | +0.63 | -1.06 | +1.34 | +0.54 |
| C3: EA/AA Names, P/U | sent | +0.52 | -0.38 | +0.73 | +1.04 | +0.65 | -0.14 | -0.30 |
| C3: EA/AA Names, P/U | c-word | NA | -0.02 | +0.93⚠️ | +0.21⚠️ | +1.05⚠️ | +0.63✅ | +1.22✅ |
| +C12: EA/AA Names, Career/Family | word | -0.15 | -0.24 | -0.58 | -0.37 | -0.95 | -1.34 | -0.87 |
| +C12: EA/AA Names, Career/Family | sent | 0.00 | -0.18 | -0.50 | -0.66 | -0.69 | -0.17 | +0.30 |
| +C12: EA/AA Names, Career/Family | c-word | NA | -0.03 | -0.09 | -0.32 | -1.09 | +0.47✅ | +0.51⚠️ |
| +C13: EA/AA Names, Science/Arts | word | -0.51 | -0.36 | -0.08 | +0.10 | +0.48 | +0.60 | +0.61 |
| +C13: EA/AA Names, Science/Arts | sent | +0.14 | -0.35 | +0.39 | -0.03 | -0.11 | +0.31 | -0.13 |
| +C13: EA/AA Names, Science/Arts | c-word | NA | +0.02 | +0.90✅ | -0.25 | +0.18 | +0.03 | -0.06 |
| +Double Bind EA/AA (Competent) | word | +1.49 | +0.22 | +0.90 | +1.20 | -0.66 | +1.21 | +0.09 |
| +Double Bind EA/AA (Competent) | sent | +1.03 | +0.14 | +1.19 | +1.05 | +0.35 | -0.30 | +0.42❌ |
| +Double Bind EA/AA (Competent) | c-word | NA | +0.10 | +0.91⚠️ | +0.31⚠️ | +0.77⚠️ | -0.81 | -0.01 |
| +Double Bind EA/AA (Competent) | sent (u) | +1.15 | -0.33 | +1.23 | +1.03 | +1.17 | -0.78 | +0.44 |
| +Double Bind EA/AA (Competent) | c-word (u) | NA | +0.06 | +1.01⚠️ | +0.70⚠️ | +0.78⚠️ | -0.70 | +0.59✅ |
| +Double Bind EA/AA (Likable) | word | +1.62 | +0.38 | +0.79 | +0.60 | -0.56 | +1.33 | +0.06 |
| +Double Bind EA/AA (Likable) | sent | +1.24 | +0.28 | +1.14 | +0.90 | -0.04 | +0.38❌ | -0.48 |
| +Double Bind EA/AA (Likable) | c-word | NA | +0.22⚠️ | +0.61⚠️ | +0.21⚠️ | +0.66✅ | -0.79 | -0.07 |
| +Double Bind EA/AA (Likable) | sent (u) | +1.29 | +0.42 | +1.30❌ | +1.02 | +0.51 | -0.53 | +0.51 |
| +Double Bind EA/AA (Likable) | c-word (u) | NA | -0.17 | -0.34 | +0.87⚠️ | -0.42 | -0.76 | -0.90 |

overall proportion of significant positive effect sizes, indicating severe pro-stereotypical bias. ELMo demonstrates the lowest proportion of significant positive effect sizes, although this might be due to the use of character convolutions for names that are out of vocabulary, which may lead to the loss of stereotypical semantics. Furthermore, BERT (bbc) exhibits the highest proportion of bias on both race and intersectional tests, and the highest proportion overall among contextual word models. We also observe that larger models tend to exhibit a smaller proportion of significant positive effect sizes. This is true for both the BERT family (blc: 0.48, bbc: 0.33) and the GPT family (GPT: 0.3, GPT-2_117M: 0.32, GPT-2_345M: 0.23). However, barring a more robust study with more model sizes in consideration, we caution against a definitive conclusion. Moreover, embedding association tests targeting race generally demonstrate a higher proportion of significant associations across models than those targeting gender, suggesting that on a broad level the problem of racial bias is more severe in word representations than gender bias. This motivates more attention on racial bias in word representations.

Furthermore, over the reported embedding association tests, we find that using contextual word representations in embedding association tests uncovers bias that using sentence encoders does not. Across the four test types and six contextual word models, we observe 93 instances where there was a significant effect either on the sent encoding or the c-word encoding. Of these 93 instances, 36.6% (34) were observed only with the c-word encoding, 25.8% (24) were observed only with the sent encoding, and 37.6% (35) were observed on both. Of the 17 embedding association tests reported in Tables 3, 4, 5, on 9 of the tests there were more positive significant associations with the c-word encoding instead of the sent encoding across all contextual word models. This indicates that when assessing social bias, the type of encoding matters determines whether the bias can be measured. In particular, contextual word representations can be used in conjunction with sentence encodings to determine bias in a given model. In Tables 3, 4, 5 and the Supplementary Material, we use colored symbols to indicate for a given model and test if significant associations were observed using either of c-word (✅) or sent (❌) encodings, or both (⚠️).

## 5.3 Analysis: Gender and Race

On embedding association tests targeting gender, we observe varying results across the models (see Table 3). Although the BERT and GPT/GPT-2 models demonstrate some significant positive effect sizes, they also unexpectedly demonstrate negative effect sizes. This seems to suggest a degree of anti-stereotypical associations, but none of the tests which have negative effect sizes are significant

**Table 5:** Intersectional embedding association tests and effect sizes, for word encodings (word), sentence encoding (sent) and contextual word representation (c-word). M/F: Male/Female. EA/AA: European American/African American. P/U: Pleasant/Unpleasant. ABW: Angry Black Woman Stereotype. Tests introduced in this paper are prefixed by "+". Gray shading indicates significant at $p = 0.01$. Green check (✓): test is significant using c-word but not sent. Red cross (✗): test is significant using sent but not c-word. Yellow triangle (⚠): test is significant using both sent and c-word.

| Test | Encoding | CBoW | ELMo | BERT (bbc) | BERT (blc) | GPT | GPT-2 (117M) | GPT-2 (345M) |
|---|---|---|---|---|---|---|---|---|
| +I1: (F) EA/AA Names, P/U | word | +1.19 | -0.01 | +1.13 | +1.43 | -1.16 | +1.07 | +0.65 |
| +I1: (F) EA/AA Names, P/U | sent | +0.15 | +0.04 | +1.35 | 0.00 | +0.44 | -0.75 | -0.75 |
| +I1: (F) EA/AA Names, P/U | c-word | NA | +0.04 | +0.98⚠ | -0.12 | +1.45⚠ | +0.09 | +0.41✓ |
| +I2: (AA) M/F Names, P/U | word | -0.63 | +0.64 | +0.96 | +1.07 | -0.78 | +0.70 | -0.49 |
| +I2: (AA) M/F Names, P/U | sent | -0.94 | +0.02 | +0.89✗ | 0.00 | -0.80 | -0.66 | -0.88 |
| +I2: (AA) M/F Names, P/U | c-word | NA | +0.07 | -0.43 | -0.10 | +0.20 | +0.31✓ | -0.23 |
| +I3: (M) EA/AA Names, P/U | word | +1.06 | -0.31 | +0.37 | +0.37 | -0.93 | +1.43 | +0.98 |
| +I3: (M) EA/AA Names, P/U | sent | +0.28 | -0.44 | +0.94 | +1.05 | +0.79 | +0.17 | +0.21 |
| +I3: (M) EA/AA Names, P/U | c-word | NA | -0.02 | +0.85⚠ | +0.43⚠ | +1.11⚠ | -0.56 | -0.49 |
| +I4: (EA) M/F Names, P/U | word | -0.22 | +0.36 | -0.42 | -0.39 | -0.48 | +1.06 | +0.21 |
| +I4: (EA) M/F Names, P/U | sent | -0.23 | -0.58 | +0.14 | -0.05 | -0.45 | +0.28 | -0.07 |
| +I4: (EA) M/F Names, P/U | c-word | NA | +0.02 | -0.59 | +0.50✓ | -0.27 | -0.31 | -0.03 |
| +I5: EA M/AA F Names, P/U | word | +0.48 | +0.48 | +1.19 | +1.26 | -1.15 | +1.64 | +0.77 |
| +I5: EA M/AA F Names, P/U | sent | -0.10 | -0.42 | +1.48 | +1.68✗ | -0.06 | -0.56 | -0.78 |
| +I5: EA M/AA F Names, P/U | c-word | NA | +0.07 | +0.42⚠ | +0.26 | +1.26✓ | -0.43 | +0.16 |
| ABW Stereotype Names | word | +1.10 | +0.53 | +1.23 | +1.69 | -0.79 | +0.87 | +0.21 |
| ABW Stereotype Names | sent | +0.62 | +0.52✗ | +1.62 | 0.00 | -0.82 | -0.70 | -0.92 |
| ABW Stereotype Names | c-word | NA | +0.19 | +1.34⚠ | +0.08 | +1.04✓ | +0.15 | -0.28 |

at $p = 0.01$ with the permutation test. The results suggest that P/U attributes are not stereotypically associated with gender, but attributes relating to work, discipline, competence and likability display gender bias. Note specifically that across BERT (bbc), GPT-2_117M and GPT-2_345M, c-word encodings often reveal bias that sent encodings do not, supporting the importance of evaluating bias at the contextual word level.

To understand the effect of gender bias propagating from the corpus level to the contextual word level, we devised test +Occ, which associates male and female names as concept words and the stereotypically male and female occupation words used in section 3 and defined in the WinoBias dataset [33]. We observe that at the c-word encoding level, the effect sizes for GPT, trained on BooksCorpus which had the lowest percentages of pro-stereotypical and anti-stereotypical occupation associations overall, are the smallest (+0.10). This is in comparison to GPT-2 (117M: +0.27, 345M: +0.43), which was trained on WebText the next least gendered corpus, and BERT, which was (bbc: +0.98, blc: +0.67) partially trained on Wikipedia the most gendered corpus. Note, however, that the result on ELMo does not support this trend (-0.27). Nevertheless, this finding reaffirms that bias tends to propagate from the corpus level to the encoding level [4].

On embedding association tests targeting race, we observe more significant positive associations, revealing an extensive problem of racial bias encoded in contextual word models (see Table 4). Although negative effect sizes are also present, similar to gender tests none of these effect sizes are significant at $p = 0.01$. The models demonstrate less evidence of racial bias on to career/family and science/art attributes than they do on attributes relating to pleasantness, competence and likability. Of note, the large BERT models (blc) and GPT models (GPT-2_345M) demonstrate few significant positive associations on gender, but many such associations on race.

### 5.4 Analysis: Intersectional Identities

On embedding association tests targeting intersectional identities (specifically the intersection of being both African American and female), we generally observe larger significant effect sizes when comparing the most privileged group against the multiple minority case (test +I5), larger than when comparing the the corresponding effect size of the most privileged group against the singular minority case of race (test +I3) or gender (test +I4) for the same encoding and model type. This is true of all instances of significant effect sizes on test +I5 except for the c-word encoding of BERT (bbc). Moreover, we find

stronger evidence for bias when comparing against race (test +I3) than when comparing against gender (test +I4). In particular, there are only 2 significant positive associations for test +I4, but there are 9 such associations for test +I3 across all models. This also coheres with the relatively high number of significant positive associations (9) on the Angry Black Woman Stereotype test.

## 6   Discussion and Limitations

This paper makes the following contributions. First, we use co-occurrence counts to show that standard corpora for pre-training contextual word models exhibit significant gender imbalances. Second, we extend existing analyses of social bias to state-of-the-art contextual word models like GPT-2, and indicate that social bias also exists in those models. This highlights the scope of the problem of fairness in state-of-the-art models for language processing. Third, we demonstrate that when measuring social bias in contextual word models, both the sentence encoding and contextual word representation should be used. It is possible that either encoding type may be unable to uncover latent social bias, whereas the other encoding type is able to. Fourth, we provide evidence for how racial bias is encoded strongly in contextual word models, potentially even more so than gender bias. Fifth, we introduce a method of comparison that anchors at the most or least privileged group to show that intersectional identities suffer from such bias as well, and more so than their constituent minority identities. In particular, we show that the effect of race on intersectional identities seems to be larger than the effect of gender.

It is important to highlight the following limitations of our work. First, the lack of significant positive associations should not be taken as an absence of social bias. Rather, this only indicate the absence as such measured by these specific tests. Second, this work assumes binary gender (male/female), which is a significant limitation in evaluating the bias of non-binary genders. We believe that the bias towards non-binary genders is likely to be worse, but there can also be more data sparsity issues in such evaluations. Third, this work only provides a preliminary investigation into the multiplicative aspects of identities of multiple minorities, in particular the specific interactions between different identities. While we have tried to isolate the effects of the different dimensions of identity in intersectional tests, more work needs to be done to determine the interactive nature of such effects.

We propose the following potential future directions. First, investigate how and why the encoding of bias may differ across both model size and model layers. Our results show that larger contextual word models seem to encode less social bias. It would be important to trace the presence of such bias across transformer or LSTM layers for each model type, to determine how bias can be hidden or potentially abstracted. Second, there can be a push for greater clarity on the types of biases encoded in a dataset. The gender skew at the corpus level can be documented similar to the datasheets proposed by Gebru et al. [12], to inform dataset users of such biases. Third, devise methods of de-biasing contextual word models. Current de-biasing methods largely address biases in context-free word embeddings [3], but the imperative for de-biasing contextual word models increases as they become more widespread.

## 7   Conclusion

This paper suggests that social and intersectional biases were not sufficiently detected with previous techniques that used sentence encodings, and suggests a new method for evaluating contextual word models at the contextual word level in order to assess social biases at both primary and intersectional levels. We acknowledge that techniques to evaluate social bias are ever evolving, and are keen to see more work on different and better methods to detect social bias in word models. Furthermore, we note that social bias detection is simply a first step. There is recent literature on methods to de-bias word embeddings, including post-processing methods [2] and techniques that use constraints during training [34, 32]. Although these methods are promising, Gonen and Goldberg [13] show that the gender bias encoded in word embeddings is mostly hidden from the defined metric of projecting onto the "he-she" vector direction. In particular, words still receive implicit gender from their associations (e.g. "receptionist" is no longer gendered with respect to "he", but is still gendered with respect to "captain"). This suggests that there is a need to expand these techniques to consider context, and our proposed use of contextual word embeddings to assess bias represents an important step in this direction. Combining the above de-biasing techniques for contextual word models remains a crucial direction for future work. Furthermore, methods for de-biasing specifically across race, gender, and intersectional identities remains a challenging open question.

## Acknowledgments

We would like to thank Jessica Ambrosio and Annique Wong for providing feedback on an early version of this paper, and the Data Science Ethics class (S&DS 150) at Yale for insightful conversations. The first author would also like to thank John Lafferty for inspiring interest in bias in word representations.

## Footnotes

[1]Extracted using https://github.com/attardi/wikiextractor on the May 4 Wikipedia dump

[2]https://github.com/openai/gpt-2-output-dataset

[3] See May et al. [21] for details on the Heilman double bind test.

[4] The full results in the Supplementary Material includes both tests using names and tests using terms.

[5] In the case of test +Occ, from the occupation words defined in the WinoBias dataset, see section 5.3

[6] https://github.com/W4ngatang/sent-bias

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
