[Supplementary Material · Appendix.pdf]

# A Details for Models

**Table 6:** Summary of models, method of encoding for each type, and representation dimension.

| Model | Sentence encoding | Contextual word encoding | Dim. |
|---|---|---|---|
| CBoW (GloVe) | mean | NA | 300 |
| ELMo | mean | Token of interest | 1024 |
| BERT (bbc) | `[CLS]` | Token of interest | 768 |
| BERT (blc) | `[CLS]` | Token of interest | 1024 |
| GPT | last token | Token of interest | 768 |
| GPT-2 (117M) | last token | Token of interest | 768 |
| GPT-2 (345M) | last token | Token of interest | 1024 |

## A.1 CBoW (GloVe)

Similar to May et al. [21], in the continuous bag of words (CBoW) model we encode sentences as the average of word embeddings using 300-dimensional GloVe vectors [7] trained on the Common Crawl corpus [24]. There is no corresponding contextual word level equivalent since GloVe is context-free.

## A.2 ELMo

Following May et al. [21], the sentence encoding of ELMo is a sequence of vectors, one for each token. We use mean-pooling over the tokens, followed by summation over aggregated layer outputs to obtain the final 1024-dimensional sentence encoding. At the contextual word level, we do not apply mean-pooling over tokens. Rather, we select the vector corresponding to the token of interest (i.e. for the sentence "This is Shanice", the vector of interest corresponds to "Shanice"), and sum over the layer outputs to obtain the 1024-dimensional contextual word encoding. We use the implementation of ELMo from AllenNLP [8].

## A.3 BERT

Similar to the original work [10], each sentence is prepended with a special `[CLS]` token and the top-most hidden state corresponding to the `[CLS]` token is used as the sentence encoding. At the contextual word level, we use the top-most hidden state corresponding to the token of interest. Since BERT uses subword tokenization, if the token of interest is subword tokenized we use the representation corresponding to the start of the token. We conduct experiments on both the 768-dimensional `bert-base-cased` (bbc) and 1024-dimensional `bert-large-cased` (blc) versions of BERT. We use the implementations of BERT from Hugging Face [9].

## A.4 GPT

Similar to May et al. [21] and the original word [26], we use the representation corresponding to the last word in the sequence as the sentence encoding. At the contextual word level, we use the representation corresponding to the token of interest. GPT also uses subword tokenization, so the start of the token is used if the token of interest is subword tokenized. Both encoding types are 768-dimensional. Different from May et al. [21], we use the implementation of GPT from Hugging Face, and not the jiant project [10].

## A.5 GPT-2

The sentence encoding and contextual word encoding is obtained exactly the same as for GPT. We conduct experiments on both the 117M version and the 345M version of GPT-2. We also use the implementations of GPT-2 from Hugging Face.

# B Details for Embedding Association Tests

The data for the embedding association tests are provided in the `data/iats` directory of the supplementary information. We build our work on the data made available by May et al. [21]. Word level tests have the

prefix `weat` but sentence level tests have the prefix `sent-weat`. The Caliskan Tests are named `weat1` to `weat10` and `sent-weat1` to `sent-weat10`. Alternate versions of the Caliskan Tests using group terms instead of names and vice versa are denotetd by the suffix `b`. The additional tests are named `weat+11` to `weat+13` and `sent-weat+11` to `sent-weat+13`, as well as `weat+i1` to `weat+i5` and `sent-weat+i1` to `sent-weat+i5`, and lastly `weat+occ` and `sent-weat+occ`. The double bind tests targeting gender are denoted with the prefix `weat_hdb` or `sent-weat_hdb`. The double bind tests targeting race are denoted with the prefix `weat_r_hdb` or `sent-weat_r_hdb`. The tests regarding the angry black woman stereotype are denoted with the prefix `weat_angry` or `sent-weat_angry`. The Caliskan Tests are detailed in Caliskan et al. [5], and the double bind tests are detailed in May et al. [21]. The additional tests are detailed in the sections below. In the full results presented in Tables 7 and 8, we also report results for neutral tests (C1-2) and tests regarding disability and age (C9-10).

In both the race and gender double bind tests, sentence templates that are either bleached of semantic meaning or unbleached are used. Following May et al. [21], examples of such templates (non-exhaustive) are as below. Target X and Y are either M/F names or EA/AA names.

**Competent, bleached**

1. **Target X**: "This is `[Name]`.", "That is `[Name]`.", "There is `[Name]`.", "Here is `[Name]`.", "`[Name]` is here.", "`[Name]` is there.", "`[Name]` is a person.", "The person's name is `[Name]`."
2. **Target Y**: "This is `[Name]`.", "That is `[Name]`.", "There is `[Name]`.", "Here is `[Name]`.", "`[Name]` is here.", "`[Name]` is there.", "`[Name]` is a person.", "The person's name is `[Name]`."
3. **Attribute A (Competent)**: "This is `[Attribute]`.", "That is `[Attribute]`.", "They are `[Attribute]`."
4. **Attribute B (Incompetent)**: "This is `[Attribute]`.", "That is `[Attribute]`.", "They are `[Attribute]`."

**Competent, unbleached**

1. **Target X**: "`[Name]` is an engineer."
2. **Target Y**: "`[Name]` is an engineer."
3. **Attribute A (Competent)**: "The engineer is `[Attribute]`."
4. **Attribute B (Incompetent)**: "The engineer is `[Attribute]`."

**Likable, bleached**

1. **Target X**: "This is `[Name]`.", "That is `[Name]`.", "There is `[Name]`.", "Here is `[Name]`.", "`[Name]` is here.", "`[Name]` is there.", "`[Name]` is a person.", "The person's name is `[Name]`."
2. **Target Y**: "This is `[Name]`.", "That is `[Name]`.", "There is `[Name]`.", "Here is `[Name]`.", "`[Name]` is here.", "`[Name]` is there.", "`[Name]` is a person.", "The person's name is `[Name]`."
3. **Attribute A (Likable)**: "This is `[Attribute]`.", "That is `[Attribute]`.", "They are `[Attribute]`."
4. **Attribute B (Unlikable)**: "This is `[Attribute]`.", "That is `[Attribute]`.", "They are `[Attribute]`."

**Competent, unbleached**

1. **Target X**: "`[Name]` is an engineer with superior technical skills."
2. **Target Y**: "`[Name]` is an engineer with superior technical skills."
3. **Attribute A (Likable)**: "The engineer is `[Attribute]`."
4. **Attribute B (Unlikable)**: "The engineer is `[Attribute]`."

## B.1 Gender

Tests targeting gender are broken down into Caliskan Tests (C6-C8b), double bind tests and additional tests (+C11). Test +C11 was created by matching the M/F names found in test C6 as targets and the P/U terms found in test C3 as attributes. Test +Occ was created by matching names from [29], tests C5 and C6 (see below under intersectional), and the stereotypical M/F occupation words from the WinoBias dataset [33].

## B.2 Race

Tests targeting race are broken down into Caliskan Tests (C3-5b), double bind tests and additional tests (+C12-13). Test +C12 was created by matching the EA/AA names found in test C3 as targets and the Career/Family terms found in test C6 as attributes. Test +C13 was created by matching the EA/AA names found in test C3 as targets and the Science/Arts terms found in test C8 as attributes. The double bind tests created for race use the EA/AA names found in test C3 as targets and the Competent/Incompetent and Likable/Unlikable terms found in the double bind tests for gender as attributes.

## B.3 Intersectional

Tests targeting intersectional identities are broken down into the angry black woman (ABW) stereotype test and additional tests (+I1-I5). For tests +I1 to +I5, we defined female European American, male African American and female African American names from [29] as well as male European American names from tests C5 and C6. The attributes used in the tests are the P/U terms found in test C3, and the targets are as follows: (+I1) female EA/AA names, (+I2) African American M/F names, (+I3) male EA/AA names, (+I4) European American M/F names, (+I5) male European American and female African American names. In this way, the tests cover a range of possible associations between different groups; in particular (+I1, +I2, +I5) anchors the basis of comparison in the most privileged group of male European American names, and (+I3, +I4, +I5) anchors comparison in the least privileged group of female African American names.

**Table 7:** Full results. Embedding association tests and effect sizes for word encodings (word), sentence encoding (sent) and contextual word representation (c-word). M/F: Male/Female. P/U: Pleasant/Unpleasant. sent(u)/c-word(u): unbleached sentence templates used. Tests introduced in this paper are prefixed by "+". Gray shading indicates significant at $p = 0.01$. Green check (✓): test is significant using c-word but not sent. Red cross (✗): test is significant using sent but not c-word. Yellow triangle (⚠): test is significant using both sent and c-word.

| Test | Encoding | CBoW | ELMo | BERT (bbc) | BERT (blc) | GPT | GPT-2 (117M) | GPT-2 (345M) |
|---|---|---|---|---|---|---|---|---|
| | | | | Neutral | | | | |
| C1: Flowers/Insects, P/U | word | +1.50 | -0.05 | -0.23 | -0.23 | +0.28 | -0.11 | +0.61 |
| C1: Flowers/Insects, P/U | sent | +1.56 | -0.05 | -0.22 | -0.25 | +0.28 | -0.11 | +0.61 ✗ |
| C1: Flowers/Insects, P/U | c-word | NA | +0.01 | +1.00 ✓ | +0.37 ✓ | +0.96 ✓ | -0.11 | +0.05 |
| C2: Instruments/Weapons, P/U | word | +1.53 | +0.39 | -0.14 | -0.12 | +1.10 | -0.59 | +0.78 |
| C2: Instruments/Weapons, P/U | sent | +1.27 | +0.43 ✗ | -0.29 | -0.30 | +1.37 ✗ | +0.90 ✗ | +0.65 ✗ |
| C2: Instruments/Weapons, P/U | c-word | NA | +0.06 | +0.78 ✓ | +0.15 | +0.14 | -0.49 | +0.13 |
| | | | | Disability, Age | | | | |
| C9: Mental/Phys, Temp/Perm | word | +1.38 | -0.52 | -0.51 | -0.27 | +1.14 | +0.65 | +0.53 |
| C9: Mental/Phys, Temp/Perm | sent | +0.39 | +0.18 | -0.71 | +0.82 | -1.20 | +0.74 | -0.61 |
| C9: Mental/Phys, Temp/Perm | c-word | NA | +0.84 ✓ | +0.38 | +0.28 | -1.39 | +0.77 ⚠ | -1.08 |
| C10: Young/Old Names, P/U | word | +1.21 | -0.47 | -0.73 | -0.57 | +0.62 | +0.96 | -0.01 |
| C10: Young/Old Names, P/U | sent | +0.50 | -0.16 | -0.39 | +0.64 | +0.69 ✗ | -0.30 | -0.39 |
| C10: Young/Old Names, P/U | c-word | NA | -0.07 | +0.14 | +0.31 | +0.17 | -0.45 | +1.05 ✓ |
| | | | | Gender | | | | |
| C6: M/F Names, Career/Family | word | +1.81 | -0.44 | -0.49 | -0.51 | -0.10 | -0.25 | -0.27 |
| C6: M/F Names, Career/Family | sent | +1.74 | -0.38 | -0.74 | -0.57 | +1.04 | +0.27 | +0.25 |
| C6: M/F Names, Career/Family | c-word | NA | -0.10 | +0.67 ✓ | -0.04 | +1.07 ⚠ | +0.39 ✓ | -0.26 |
| C6b: M/F Terms, Career/Family | word | +0.55 | -0.22 | -0.42 | +0.09 | +0.04 | +0.30 | +0.38 |
| C6b: M/F Terms, Career/Family | sent | +0.40 | -0.34 | -0.39 | +0.23 | +0.45 ✗ | +0.17 | +0.25 |
| C6b: M/F Terms, Career/Family | c-word | NA | -0.07 | +0.09 | -0.17 | +0.27 | +0.36 ✓ | +0.32 |
| C7: Math/Arts, M/F Terms | word | +1.06 | -0.44 | +0.04 | -0.09 | +0.58 | +0.21 | +0.24 |
| C7: Math/Arts, M/F Terms | sent | +0.77 | -0.48 | -0.10 | -0.33 | +0.82 | +0.22 | -0.63 |
| C7: Math/Arts, M/F Terms | c-word | NA | -0.10 | +0.98 ✓ | +0.42 ✓ | +0.85 ⚠ | +0.11 | +0.55 ✓ |
| C7b: Math/Arts, M/F Names | word | +1.61 | +0.15 | +0.08 | +0.41 | +0.40 | +0.32 | +0.43 |
| C7b: Math/Arts, M/F Names | sent | +1.51 | +0.02 | -0.30 | -0.05 | +1.25 | +0.23 | -0.04 |
| C7b: Math/Arts, M/F Names | c-word | NA | -0.09 | +1.29 ✓ | +0.31 | +0.79 ⚠ | +0.05 | -0.36 |
| C8: Science/Arts, M/F Terms | word | +1.24 | +0.24 | -0.23 | -0.15 | +0.25 | +0.51 | +0.87 |
| C8: Science/Arts, M/F Terms | sent | +1.01 | -0.30 | +0.11 | -0.16 | +0.89 | -0.15 | -0.15 |
| C8: Science/Arts, M/F Terms | c-word | NA | +0.16 | +1.02 ✓ | -0.08 | +1.03 ⚠ | +0.64 ✓ | +0.67 ✓ |
| C8b: Science/Arts, M/F Names | word | +1.52 | +0.36 | +0.30 | +0.46 | -0.01 | +0.54 | -0.10 |
| C8b: Science/Arts, M/F Names | sent | +1.39 | +0.55 ✗ | +0.12 | +0.11 | +0.77 | -0.23 | +0.24 |
| C8b: Science/Arts, M/F Names | c-word | NA | +0.04 | +0.72 ✓ | -0.03 | +0.81 ⚠ | +0.24 | -0.23 |
| +C11: M/F Names, P/U | word | -1.31 | +0.34 | +0.69 | +0.83 | -0.43 | +0.82 | -0.10 |
| +C11: M/F Names, P/U | sent | -0.87 | +0.15 | +0.68 ✗ | +0.18 | -0.64 | +0.27 | -0.17 |
| +C11: M/F Names, P/U | c-word | NA | +0.14 | -0.44 | +0.27 | -0.35 | +0.46 ✓ | -0.13 |
| +Occ: M/F Names, Occ Terms | word | +1.59 | +0.63 | +0.55 | +0.65 | -0.38 | +0.76 | +0.46 |
| +Occ: M/F Names, Occ Terms | sent | +1.48 | +0.06 | +0.30 | +0.51 | +1.74 ✗ | -0.00 | -0.27 |
| +Occ: M/F Names, Occ Terms | c-word | NA | -0.27 | +0.98 ⚠ | +0.67 ⚠ | +0.10 | +0.27 ✓ | +0.43 ✓ |
| Double Bind M/F (Competent) | word | +1.62 | -0.34 | -0.35 | -0.26 | -0.66 | +1.00 | -0.04 |
| Double Bind M/F (Competent) | sent | +0.79 | -0.15 | -0.06 | 0.00 | +0.27 | +0.52 ✗ | +0.25 |
| Double Bind M/F (Competent) | c-word | NA | -0.07 | +0.42 ✓ | +0.02 | -0.02 | -0.94 | +0.57 ✓ |
| Double Bind M/F (Competent) | sent (u) | +0.84 | +0.21 | +0.39 | +0.60 | -0.76 | +1.26 ✗ | -0.59 |
| Double Bind M/F (Competent) | c-word (u) | NA | -0.48 | +0.46 | -0.37 | -0.36 | -0.72 | +0.56 |
| Double Bind M/F (Likable) | word | +1.29 | -0.61 | -1.37 | -0.64 | +0.15 | +0.83 | +0.02 |
| Double Bind M/F (Likable) | sent | +0.69 | -0.45 | -0.66 | -0.29 | -0.53 | -0.44 | -0.13 |
| Double Bind M/F (Likable) | c-word | NA | -0.38 | +0.64 ✓ | +0.13 | -0.03 | -0.68 | +0.50 ✓ |
| Double Bind M/F (Likable) | sent (u) | +0.51 | -0.92 | +0.74 | -0.97 | -1.57 | +0.25 | -1.01 |
| Double Bind M/F (Likable) | c-word (u) | NA | +0.20 | +1.29 ✓ | -0.78 | -1.22 | -0.98 | +0.39 |

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

## Footnotes

[7]Downloaded from https://nlp.stanford.edu/projects/glove/

[8]https://github.com/allenai/allennlp/blob/master/tutorials/how_to/elmo.md

[9]https://github.com/huggingface/transformers

[10]https://github.com/nyu-mll/jiant