[Reviews · NeurIPS 2019]

Reviewer 1



[Update: thank you for providing the author response. I look forward to the final version including more details about the tests, as requested by reviewer 2.] This paper studies the presence of social biases in contextualized word representations. First, word co-occurnce statistics of pronouns and stereotypical occupations are provided for various datasets used for training contextualizers. Then, the word/sentence embedding association test is extended for the contextual case. Using templates, instead of aggregating over word representations (in sentence test) or taking the context-free word embedding (in word test), the contextual word representation is used. Then, an association test compares the association between a concept and an attribute using a permutation test. The number of significant associations across tests is recorded for every word representation model. The results show that some representation exhibit more bias in more tests than others. Some of the interesting trends are that larger models exhibit fewer positive effects and that contextualized word representations present different biases than sentence encoders. Race bias seems to be more pronounced than gender bias, both on its own and when intersected with gender. The discussion acknowledges certain limitations of the work in representing gender and more nuanced identities, and points to directions for future work. Originality: the work is a natural next step after the investigation of bias in context-free embeddings and in sentence representations. The method is a simple extension from WEAT/SEAT. The contribution is well situated w.r.t prior work. Quality: the paper presents a nice set of experiments that explore different possible manifestations of bias. The appendix provides more complete results. The appear also acknowledges and discusses potential limitations. Some further analysis would be welcome, such as: - Relate the overall results in section 4.5 to the corpora statistics in table 1. - Clarify and expand the discussion of overlap or no overlap between tests revealed by contextualized representations and sentence encoders (sec 4.5, second paragraph). It's a bit hard to follow. - Test the claim that bigger models exhibit less bias by training models of increasing size (but otherwise identical), e.g. Glove with different sizes. - More discussion of the intersectional results in 4.7. What do they mean? Clarity: the paper is well written and very clear. A few minor comments: - Define intersectional earlier. - Define acronyms on first occurrence (e.g. LSTM) - corpuses -> corpora? - Say a bit more on the counting method from [37]. - line 113: table 1. Also refer to other tables from the text. - Table 2: give total number of tests per type (say, how many race tests are there in total). On the other hand, the current "total" column is misleading because it counts the same tests multiple times (across models). - Line 130: sentence seems broken. - Line 142: the the - Section 4.2: line 153 says "we use the contextual word representation of the token of interest, before any pooling is used to obtain the sentence encoding". If I understand correctly, the contextual representation is used instead of pooling; that is, in this case, there is no pooling. Right? The specific method should be described in the main paper in my opinion. - line 259: onto (a) the Significance: The results are important for the NLP community, where both contextualized word representations and bias are topics of much research. The results show complementary benefits to previous work on sentence encodings (although this part can be analyzed better).

Reviewer 2



UPDATE AFTER READING RESPONSE: Many of my concerns have been addressed; thank you for the careful response! Some minor fixes: - line 128-130 sentence is messed up (noted in another review) - Eq 3 has a typo - missing "s(" I think - line 136: "a more severe pro-stereotypical" grammar / word choice error. Maybe they mean "pro-stereotypical representation"? ============== This work tests contextual neural network language representations for social biases in their representations of people, for stereotypes associated with race, gender, and their intersection. This is a nice advance to this fairly new, rapidly growing, and important literature on social biases in AI systems, that has very deep connections to regularities in human cognition, social psychology, language use, and huge implications for machine learned AI systems. The paper finds substantial biases persisting across a range of the latest-and-greatest models, and in particular, within the contextually aware models (ELMo/LSTMs, BERT/GPT/self-attention) that recently supplanted acontextual word embeddings in current NLP research. Right now, I believe there are two competing frameworks for testing bias in word embeddings. Bolukbasi's linear projection analysis was, in my opinion, fairly convincingly critiqued by Gonen and Goldberg at NAACL this year; fortunately, this paper uses the understudied (in NLP/ML) criterion from Calisken, which compares within- versus between- pairwise simlarities of words in two groups. The authors argue that the previously proposed method to analyze contextual models - May et al.'s "SEAT" -- was insufficient. I agree it seems like that, but the proposal in this work - to use the top layer of per-token embeddings - seems like an obvious step. It doesn't say this work is a big advance that it's better than May et al.; it says May et al. was oddly limited. But still, the test is insufficiently explained, even after reading the appendix. The Calisken test (eq 1-4) is presented in terms of averages among "concept embeddings." But the tests are defined by wordlists. Is there averaging among a word type's instances in the corpus? Or are token-to-token similarities used to calculate the distances? This work re-examines a number of the wordlist tests from previous work on thes newer models, and proposes a few new wordlists tests as well. I wish the new wordlist tests were explained more; it is a difficult socio/cultural/linguistic problem to create a robust and social scientifically valid wordlist to draw conclusions from. For example, for the new or modified versions of the Heilman double bind -- a nontrivial phenomenon -- what exactly was done for " we also extend the attribute lists of competence and likability to the context of race. "? In general, since there was heavy reliance on May et al., I found it a little hard to follow what exactly was being done in parts of this paper. The comparison of race versus gender bias was interesting, though I wish there had been more. I don't love the approach of counting significant effects. Please see the ASA's critique of p-values from 2016 ("Statement on Statistical Significance and P-Values"). Effect size may be an important additional way to interpret the meaning of the differences among these models. Or maybe I'm wrong and there's a better way to analyze the data out there. I'm not convinced what's done in this paper is a great way to do it. The results seem like a straightforward or obvious documentation, without much insight about what's going on. For example, did any of the previous studies find negative effects, or are these examples for BERT/GPT the very first time it's been observed? What does it mean? How do findings on intersectional identities relate to the social science/scholarship literature on them? This style of paper and analysis is potentially very interesting, and can be a huge contribution to the field. For example, in terms of one specific finding, I found myself impressed with the Calisken paper's connection of real-world employment data all the way to what word embeddings learn; it's a nice illustration of how AI systems are deeply embedded in both concrete and stereotypical facts about society. To make this type of contribution, the discussion and analysis is key, beyond reporting numeric results, to get into the social and AI system hypotheses the work has implications for. As a technical issue, context could be quite interesting (e.g. do certain contexts mediate or affect bias?), and as a social issue, intersectionality could be fascinating. Unfortuantely, the interesting aspects of this area don't come through strongly in this paper. The work has lots of potential, and I hope the authors can improve or explain it better to get there.

Reviewer 3



The authors use known asociation tests as the basis for their analysis and include existing measures and bleached sentences. Their primary contribution is the addition of race and a different approach for analyzing contextualized word embeddings (not moving to the sentence level). They find slight but predictable correlations on known biases across the population splits they analyze and differences between both model sizes and datasets. This includes the surprising result that larger models appear to exhibit less of the known issues than expected.

[Author Response · NeurIPS 2019]

**The authors sincerely thank the reviewers for their time and effort; your remarks will help us improve the quality of our paper, and we respond to each reviewer's queries individually below.**

**Reviewer 1:**  Thank you for the detailed thought-provoking comments which will help us improve the final work.

**Overlap.** Some tests reveal bias only when c-word encoding is used, and other tests reveal bias only when sent encoding is used; this suggests that a single metric does not suffice and that our method exposes biases which may otherwise be missed. We will include a detailed illustration of the overlaps (showing which tests exhibit bias in which models) in the final version.

**Larger models exhibit less bias?** We report two BERT model sizes (bbc_110M vs blc_370M) and the GPT model family (GPT, GPT-2_117M and GPT-2_345M). This hypothesis is supported by the number of significant positive effect sizes (Table 2), however we note that the effect sizes and significance vary in both directions across specific tests (Tables 3, 4, 5), so we do not believe we can conclude that bias "reduces" as it appears to change in nature. Because of how expensive (and inaccessible) it is to pre-train these models, we are unable to conduct a more robust study on model size and bias effect size for context dependent models. Evaluating GloVe (50d, 100d, 200d and 300d under the CBoW setup in our paper), we found no consistent trend across embedding size on effect size for either word or sent encodings.

**Corpora statistics and results.** We will include a test for associating M/F names and occ words in the final version.

**Section 4.2: line 153.** Indeed the method uses the contextual representation, so there is no pooling involved.

**Reviewer 2:**  Thank you for the incisive comments which help us improve our discussion and interpretation of results.

**Tests.** We define concepts to be notions of classes like M/F and EA/AA, and define attributes to be characteristics that can be assigned to these classes, such as P/U and Career/Family. Each concept and attribute is defined with a word list (or sentence list), and thus we do not use any corpus or averaging. Regarding token-to-token similarities, specifically we calculate cosine similarities between word/sentence/contextual word encodings.

**New word lists.** Indeed, our word lists were constructed in prior work (Caliskan et al. [6] and May et al. [23]), which grounded the lists in the social science literature. Our contribution is in exploring them more fully across different permutations and with contextual models. E.g., for the extension of the Heilman double bind tests to race, we kept the same attribute word lists as the original tests, but replaced M/F names with EA/AA names (see also Appendix B.1-B.3).

**Counting significant effects.** We include Table 2 for ease of interpretation, however our main analysis is in the form of effect sizes (Tables 3-8).

**Negative effect sizes.** Some prior work [23] also found negative effect sizes for BERT and GPT (for sent encodings). While surprising, note that none of the instances of negative effect sizes we observed were found to be significant given the permutation test.

**Intersectional results.** We find that a similar proportion of our intersectional tests exhibit significant positive effects as our tests on race ( 25%); gender tests have a smaller proportion ( 12%). The experience of multiple minorities is at least as worse as their constituent minorities, but based on [12] we expected larger effect sizes in our intersectional tests. For this response, we further compared a test on M EA/F AA names with M EA/M AA and M EA/F EA names, finding that BERT exhibits larger significant effect sizes on the multiple minority case (1.57) than the others (0.68, 1.21).

**Reviewer 3:**  Thank you for your helpful comments which will help us improving the exposition of our results.

**1.** In Tables 2 and 3, we do find that the c-word encoding on the non-double bind tests in general have higher effect sizes than with the sent encoding. We believe this is due to the modulating effect of pooling operations (ELMo) or the use of first (BERT) / last (GPT) word representations to obtain sent encodings.

**2.** Indeed, "they" is primarily used as the collective pronoun in these corpora; the takeaway observation from Table 1 is that it has more M-biased occurrences than F-biased occurrences despite being theoretically neutral.

**3.** We agree that terms relating to assistive devices would make sense to include. However, we determined that developing new lists was outside the scope of our expertise as it should be carefully grounded in the social science literature, and hence we only used lists developed and vetted by other scholars (e.g., [6] for ableism and age).

**4.** This is an interesting suggestion which we have not seen in the existing literature. It could be attained by considering negative examples (sentences which "should" be bias neutral given our understanding from the social sciences) to see if correlations are still observed. Developing such lists would be an interesting cross-disciplinary challenge.

**5.** There is no clear "best" method for measuring bias in any domain; indeed, it is unlikely that any single test will suffice. Rather, this work suggests that our method captures aspects of bias that other tests fail to discern.

[Meta-Review · NeurIPS 2019]

This paper addresses social bias in contextualized word representations, a timely and important problem. The authors here find that in contextualized embeddings yielded from modern neural LMs, biases persist. This is an important finding. Reviewers expressed some concerns regarding the experimental setup and metrics, but felt these were adequately addressed by the author response.